# Conversion of Bivalve Shells to Monocalcium and Tricalcium Phosphates: An Approach to Recycle Seafood Wastes

**DOI:** 10.3390/ma14164395

**Published:** 2021-08-05

**Authors:** Somkiat Seesanong, Banjong Boonchom, Kittichai Chaiseeda, Wimonmat Boonmee, Nongnuch Laohavisuti

**Affiliations:** 1Department of Plant Production Technology, Faculty of Agricultural Technology, King Mongkut’s Institute of Technology Ladkrabang, Bangkok 10520, Thailand; ksesomki@yahoo.com; 2Advanced Functional Phosphate Material Research Unit, Department of Chemistry, Faculty of Science, King Mongkut’s Institute of Technology Ladkrabang, Bangkok 10520, Thailand; 3Municipal Waste and Wastewater Management Learning Center, Faculty of Science, King Mongkut’s Institute of Technology Ladkrabang, Bangkok 10520, Thailand; 4Organic Synthesis, Electrochemistry and Natural Product Research Unit (OSEN), Department of Chemistry, Faculty of Science, King Mongkut’s University of Technology Thonburi, Bangkok 10140, Thailand; 5Department of Biology, Faculty of Science, King Mongkut’s Institute of Technology Ladkrabang, Bangkok 10520, Thailand; bwimonmat@gmail.com; 6Department of Animal Production Technology and Fishery, Faculty of Agricultural Technology, King Mongkut’s Institute of Technology Ladkrabang, Bangkok 10520, Thailand; nongnuch.la@kmitl.ac.th

**Keywords:** calcium phosphate, calcium carbonate, recycling, environmental problems, seashell

## Abstract

The search for sustainable resources remains a subject of global interest and the conversion of the abundantly available bivalve shell wastes to advanced materials is an intriguing method. By grinding, calcium carbonate (CaCO_3_) powder was obtained from each shell of bivalves (cockle, mussel, and oyster) as revealed by FTIR and XRD results. Each individual shell powder was reacted with H_3_PO_4_ and H_2_O to prepare Ca(H_2_PO_4_)_2_·H_2_O giving an anorthic crystal structure. The calcination of the mixture of each shell powder and its produced Ca(H_2_PO_4_)_2_·H_2_O, at 900 °C for 3 h, resulted in rhombohedral crystal β-Ca_3_(PO_4_)_2_ powder. The FTIR and XRD data of the CaCO_3_, Ca(H_2_PO_4_)_2_·H_2_O, and Ca_3_(PO_4_)_2_ prepared from each shell powder are quite similar, showing no impurities. The thermal behaviors of CaCO_3_ and Ca(H_2_PO_4_)_2_·H_2_O produced from each shell were slightly different. However, particle sizes and morphologies of the same products obtained from different shells were slightly different—but those are significantly different for the kind of the obtained products. Overall, the products (CaCO_3_, Ca(H_2_PO_4_)_2_·H_2_O, and Ca_3_(PO_4_)_2_) were obtained from the bivalve shell wastes by a rapidly simple, environmentally benign, and low-cost approach, which shows huge potential in many industries providing both economic and ecological benefits.

## 1. Introduction

Seafood productions contributed importantly to require for a source of protein worldwide, but it also creates huge waste quantities of solid and liquid in the processes [1]. For mollusk, shell wastes of over 13 million tons were manufactured yearly [2,3]. Three major types of mollusk informed by the Food and Agriculture Organization of the United Nations (FAO) Fisheries and Aquaculture Department are cockles, mussels, and oysters, which are consumed very largely around the world [1]. The main sources are generally from aquaculture upward than wild fisheries, amongst which oysters were dominant followed by mussels and cockles. In 2018, 5.8, 1.6, and 0.4 million tons of oyster, mussel, and cockle were produced, respectively [2]. Generally, bivalve shell wastes account for about 65–80% of live weight, which is expected to be over 5 million tons a year [4]. Large numbers of bivalve shells are dumped into public waters and/or landfills and create numerous environmental obstacles that contribute to pollution to coastal fisheries, public water surface, an unpleasant smell as a consequence of the decomposition of organics attached to the shells, and natural landscape affecting to health/sanitation problems [2,3,4,5]. Consequently, the disposal of bivalve shell wastes is getting an extremely fatal issue for the marine aquaculture industries and various consumer countries. The increasing knowledge of sustainable evolution and research attention in innovative technologies on the conversion of bivalve shell wastes into helpful and expensive chemicals and compounds have been starting in the 21st century [4]. So far, many researchers have studied on characterizations of bivalve (cockles, mussels, and oysters) shells and reported that chemical contents consist of primarily calcium carbonate (>95%) with various crystal phases [4]. Based on their environment, the different species of shells may comprise various quantities of cation contaminations such as silicon, magnesium, aluminum, strontium, phosphorus, sodium, or sulfur [6,7,8].

Calcium carbonate (CaCO_3_) naturally occurs in rocks and shells of various organisms and is widely used in construction, papermaking, pharmaceuticals, agriculture, etc. This CaCO_3_ compound occurs naturally in three polymorphs including calcite (β), aragonite (γ), and vaterite (μ) [6,7,8]. Nowadays, the principal calcium carbonate production is from mineral resources, which have the risk of heavy metal contamination and are non-renewable resources, unlike calcium carbonate from bio-derived shells which are generally abundant, renewable, inexpensive, and environmentally friendly [1,4]. Based on the above mentioned, recycling seashell wastes to CaCO_3_ raw material offers many advantages and has potential application in various fields (Figure 1). Various worldwide research shows immense potential for applications of seashells. Recently, they have been used to produce hydroxyapatite [7], nano-hydroxyapatite (nHA) [8], apatite nanoparticles [9], calcite lime [10], CaO [11,12], bio-filler in polypropylene [13], matte glaze [14], cement clinker [15], cementitious construction materials [16], expansive additive in cement mortar [17], adsorbent for Pb(II) adsorption [18], adsorbent for sulfate and metals removal [19], covalently functionalized biogenic CaCO_3_ [20], calcined mussel shell powder (CMSP) for antistatic oil-removal [21]. However, in Southeast Asia, especially Thailand, a country with the highest bivalve (cockles, mussels, and oysters) production, the waste shell recycling means is not created appropriately, and these wastes are mainly dumped in the near areas affecting an environmental issue [22,23]. Alarmed with the problems, the Ministry of Higher Education, Science, Research, and Innovation of Thailand planned to resolve according to the Bio-Circular-Green Economy (BCG) model and financed a program to set new strategies for recycling these wastes, including establishing factories for producing calcium compounds to increase the recycling quantity of bivalve shell wastes. However, only 30% of bivalve shell wastes are reused/recycled by these factories [24,25].

Realizing this problem, our research focuses on the conversion of bivalve shells to calcium phosphates, which are used as nutritional supplements, catalysts for some chemical reactions, fertilizers, and animal feed minerals, the mineral basis of the tooth and bone tissues, as well as for creating materials with unique properties [1,4,5,6,7,8,9]. In Thailand, monocalcium phosphate (MCP) and tricalcium phosphate (TCP) are enormously used in many fields and both compounds are imported every year. MCP has been used in huge quantities in agriculture. It is called superphosphate fertilizer, classified by three levels of %P_2_O_5_ (single (9–20%), double (20–48%), and triple (48–58%) superphosphates), and P-21 for animal feed minerals [24,25]. Additionally, MCP is also used in large amounts in the food industry as a buffer, hardener, leavening reagent, yeast food, beverage, bakery, and nutrient [26]. TCP is widely used in huge amounts in the medical and pharmaceutical industries as medicine, tooth, bone, calcium supplement, in the animal feed industry as calcium additive and supplement, and in the food industry as various functions (acidity regulator, anticaking, emulsifying, firming, flouring, humectant, raising, stabilizer, and thickener) in many food substances under the number 341(iii) [27]. Both compounds have been synthesized to produce high-purity grades from various calcium compounds (chloride, carbonate, oxide, nitrate, acetate, etc.) and various phosphorus compounds (phosphoric acid, sodium, potassium, ammonium, etc.) by many methods including chemical precipitation [24], hydrothermal synthesis [28,29], microwave-assisted methods [30,31], precipitation of emulsions [32], sol-gel [33], crystallization of solutions [34], chemical deposition [35], electrodeposition [36], and mechanic-chemical synthesis [37]. The method performed in an individual event depends on the requisite kind of morphology, structure, and chemical content. However, the typical drawback of these synthesized methods is expensive raw materials resulting in the high cost of the obtained products which are not suitable to use for some applications such as fertilizer and animal feed industries.

Therefore, one of the solving keys for the mention-above points is to apply/recycle the bivalve shell wastes in such a route that it can be more valuable and resourcefully used to create these calcium phosphates which may solve some financial issues to purchase expensive compounds for many industries in Thailand. Although, MCP and TCP have been reportedly prepared from bivalve shell wastes such as oyster shells [38], and Mediterranean mussel shells [39], the methods used complex and high-cost processes which many parameters must be carefully controlled (concentration, pH, time, and temperature). The aim of this present work is to easily and quickly obtain calcium carbonate from bivalve shell wastes (cockles, mussels, oysters) and then subsequently use it to produce MCP and TCP by using an easy, cost-effective, and environmentally benign method. Moreover, this work also highlights some potential applications for shell wastes that can bring both economic and ecological benefits.

## 2. Materials and Methods

### 2.1. Starting Reagents

Raw reagents utilized for the current study were waste bivalve shells of cockle, mussel, and oyster collected from seafood restaurants of fishermen residing in areas of the Chonburi beaches in eastern Thailand. The individual kind of seashells derived in the primitive shape was carefully cleaned with triply distilled water and dried in an oven at 100 °C for 3 h. Each kind of dried seashell was pulverized to produce fine powder by using an agate mortar and pestle and then was sieved in 100 mesh (150 μm). All fine seashell powders were then characterized to identify the purity and solid phase of calcium carbonate before proceeding to the preparation of the calcium phosphates. Fine seashell powders of cockle, mussel, and oyster were CaCO_3_ compounds and denoted with the sample codes CSP, MSP, and OSP, respectively.

### 2.2. Monocalcium Phosphate Monohydrate (Ca(H_2_PO_4_)_2_·H_2_O, MCPM) Preparation

A collection of monocalcium phosphate hydrate samples was synthesized by the reaction of individual seashell powder (cockle (CSP), mussel (MSP), and oyster (OSP) shells) with 85 wt% phosphoric acid and distilled water, using a constant-addition method modified from the generic reaction reported in previous work [10].

Briefly, 15 mL of 85 wt% H_3_PO_4_ was slowly added into a beaker (100 mL), which contains 9 g of CSP, and was constantly stirred with a Teflon stir bar. This mixing reaction exhibited an exothermic process noticed by increasing temperature (65 °C). Then, 12 mL of distilled water (H_2_O) was immediately added into the resulting mixture with constant stirring until CO_2_ gas bubbles are no evolved (about 30 min). The resulting reaction has been stayed in the open air for about 3 h to become dried powder of monocalcium phosphate hydrate without other processes such as filtration and drying with temperature control. The monocalcium phosphate hydrate obtained from CSP was labeled with the sample code MCP-C. For MSP and OSP, the processes were repeated in the same way as CSP, and the obtained products were labeled as MCP-M and MCP-O, respectively.

### 2.3. Tricalcium Phosphate Anhydrous (Ca_3_(PO_4_)_2_, TCP) Preparation

A series of tricalcium phosphate samples was prepared by mixing powders of individual seashell powder (cockle (CSP), mussel (MSP), and oyster (OSP) shells) with their prepared monocalcium phosphate hydrate pair (MCP-C, MCP-M, and MCP-O). The generic reaction is:(1)$$2{\mathrm{CaCO}}_{3}(s)+\mathrm{Ca}(H_{2}{\mathrm{PO}}_{4}{)}_{2}\cdot H_{2}O(s)\to {\mathrm{Ca}}_{3}({\mathrm{PO}}_{3}{)}_{2}(s)+H_{2}O(g)$$


In the typical way, 2.0 g of CSP and 2.52 g of MCP-C were weighed and then well mixing by grinding in a crucible. Then, the mixed powder was calcined at 900 °C for 3 h in a furnace. Its final product after heating is tricalcium phosphate, labeled as TCP-C. The tricalcium phosphate powders prepared from the mixed powders of MSP + MCP-M and OSP + MCP-O were prepared in the same way as CSP + MCP-C and the obtained products were labeled as TCP-M and TCP-O, respectively.

### 2.4. Sample Characterization

#### 2.4.1. Thermogravimetric Analysis (TA)

The thermal behaviors of the dried fine seashell powders and monocalcium phosphate monohydrate were analyzed by a thermogravimetric/differential thermal analyzer (TG-DTA, Pyris Diamond, Perkin Elmer). The experiments were performed in the static air, at the heating rates of 10 °C min^−1^ over the temperature range from 30 to 900 °C and the O_2_ flow rate of 100 mL min^−1^. The sample mass of about 6.0–10.0 mg was filled into an alumina crucible without pressing. The thermogram of a sample was recorded in an open aluminum pan using Al_2_O_3_ as the reference material.

#### 2.4.2. Fourier Transform Infrared (FTIR) Spectroscopy

The molecular structures were measured by a Fourier Transform Infrared Spectrophotometer (FTIR, Spectrum GX, Perkin Elmer), which were recorded in the range of 4000–400 cm^−1^ with eight scans and the resolution of 4 cm^−1^ using KBr pellets (spectroscopy grade, Merck).

#### 2.4.3. Powder X-ray Diffraction (XRD)

The structure of the prepared samples was recorded by X-ray powder diffraction using an X-ray diffractometer (D8 Advance, Bruker AXS GmbH) with Cu K radiation (λ = 0.1546 nm) operating at the condition of 40 kV and 40 mA. The specimen was pulverized into a fine powder and used for the analysis. The diffraction angle was continuously scanned from 10° to 60° in 2θ at a scanning rate of 2°/min. A range of 10–60° is shown in the figures because no relevant peaks occurred in the excluded region.

#### 2.4.4. Scanning Electron Microscopy (SEM)

The morphology of the selected resulting samples was determined by a scanning electron microscope using LEO SEM VP1450 after gold coating.

## 3. Results and Discussion

### 3.1. Characterization Results of Bivalve Shell Powders

Figure 2 displays TG/DTG curves of the CSP, MSP, and OSP samples, which are quite similar. TG lines of each sample show the mass loss in the region of 600–800 °C, which correspond to a strong single peak of DTG curves at 752, 772, 750 °C for the CSP, MSP, and OSP samples, respectively. Four DTG peaks observed at 515, 540, 569, and 625 °C for the MSP sample may have resulted from other cation contaminations that may be formed the mixing phase of metal carbonates, which can be not pointed out still clearly. The quantities of mass loss are found to be 43.1% for the CSP sample, 43.9% for the OSP sample, and 48.4% for the MSP sample. The thermal results were well consistent with those of the reference data of CaCO_3_ and theoretical data [17,40,41]. The thermal behavior obtained indicates that the bivalve shell powders can be transformed to CaO by calcination at above 772 °C, which may be useful for the production of this compound to be used in specific applications.

Figure 3 illustrates FTIR spectra of the CSP, MSP, and OSP samples that are quite similar due to fundamental vibrational bands of CO_3_^2−^ block unit in the CaCO_3_ structure for each sample. The vibrational modes of the CO_3_^2−^ anion are divided into three types [42]: (i) internal vibrational modes of (CO_3_^2−^) groups, (ii) hydroxyl vibrations (in the case of hydroxyl carbonates ≈ 900 cm^−1^, 1500–1600 cm^−1^, and 3400 cm^−1^), and (iii) vibrational M-O modes from the interactions between the cation and oxygen of either (CO_3_^2−^) or OH^−^ (external or lattice modes). The carbonate anion (CO_3_^2−^) is a nonlinear structure with four atoms resulting in six (3 × 4 − 6) normal modes of vibrations [43]. The six normal vibrational modes are a nondegenerate symmetric stretch (*v*_1_; A’_1_: Raman active), nondegenerate asymmetric (out of plane) bend (*v*_2_; A”_1_: IR active), doubly degenerate asymmetric stretch (*v*_3_; E’: Raman and IR active), and doubly degenerate symmetric (in-plane) bend (*v*_4_; E’: Raman and IR active). The FTIR spectra of the CSP, MSP, and OSP samples were analyzed according to this theory. Two strong intense bands at 696 cm^−1^ and 863 cm^−1^ are assigned to the *ν*_4_ and *ν*_2_ modes, respectively. A weak band at 1030 cm^−1^ is contributed to *v*_1_ mode. A band at 1413 cm^−1^, looking like a mountain, is related to *v*_3_ mode. A weak band observed at 1782 cm^−1^ may be respected as the combination bands of *ν*_4_ + *ν*_1_ modes. A weak band at 2520 cm^−1^ and a broad band around 2875 cm^−1^ may be regarded as a combination of or/and overtone of *ν*_4_, *ν*_3_, and *ν*_1_ modes. A single band at 3453 cm^−1^ was assigned to the OH-stretching modes. For the FTIR results of all bivalve shells, the *v*_1_ mode that appear normally active in Raman is observed and *v*_3_ and *v*_4_ modes are not shown doubly degenerate bands, which may be correlated with the atomic cation masses and the presence of molecules belonging to site symmetry of their structures [44]. The FTIR results obtained are very similar to that of the calcite phase of CaCO_3_ in literature [43,44], which indicates that the CSP, MSP, and OSP samples have the main content as this crystalline phase.

XRD patterns of the CSP, MSP, and OSP samples are very similar and are exhibited in Figure 4. Bivalve (cockle, mussel, and oyster) shells, biological wastes are primarily made up of calcium carbonate (≥96wt%) and little contaminations of other chemical contents. It can clearly indicate that the main crystalline phases of the CSP, MSP, and OSP samples are calcite (β-CaCO_3_) and minor aragonite are detected for OSP while little vaterite is detected for the other two samples, shown in Figure 4. The diffraction intensity is the calcite (002) 2θ = 29.81 and the next to strongest are the calcites at (111), (012), (202), (112) (200), and (202), respectively. The XRD analysis verified that the crystalline phase of CaCO_3_ in the CSP, MSP, and OSP samples was calcitic polycrystals, which was found to match with the PDF data file of CaCO_3_ (PDF no.72–1937) [41,45]. The XRD results are in well agreement with the FTIR data.

SEM micrographs of the CSP, MSP, and OSP samples exhibit different morphological features and are shown in Figure 5. SEM image of CSP reveals an elongated, partly polyhedral morphology of rod-like crystals (up to 10 µm long) with different sizes and forms. SEM image of MSP reveals elongated, partly polyhedral morphology of sheet-like crystals with different sizes (5–10 μm). Finally, the SEM image of OSP reveals plate-like crystals of different sizes, which are agglomerated.

### 3.2. Characterization Results of Monocalcium Phosphates

TG/DTG curves of three monocalcium phosphate samples prepared from cockle (CSP), mussel (MSP), and oyster (OSP) shell powders, labeled as MCP-C, MCP-M, and MCP-O, respectively, are displayed in Figure 6. TG lines of all samples showing the mass loss in the range of 100–600 °C are similar. The mass losses found to be around 20% for each sample correspond to the loss of three water molecules in structure. Total mass losses found to be 18.9% for MCP-C, 25.0% for MCP-M and 17.3% for MCP-O are slightly different from that of theoretical value at 21.4% [41,46]. The obtained results indicate that the number of water molecules in Ca(H_2_PO_4_)_2_·nH_2_O structure would be 0.7, 1.5, and 0.4 mole for the MCP-C, MCP-M, and MCP-O samples, respectively. The number of crystal water found in the range of 0-n < 2 is an impossible theory, which is consistent with that of the previous works [24,26,41,46]. The relative with TG data, DTG curves of the MCP-C, MCP-M, and MCP-O samples showing the numbers and peak positions of steps of thermal transformation are different. Four DTG peaks are observed at 127, 178, 224, and 330 °C for the MCP-C sample and 127, 185, 245, and 330 °C for the MCP-M sample while five DTG peaks occur at 127, 185, 224, 265, and 330 °C for the MCP-O sample. Two peaks that occurred below 200 °C correspond to the dehydration steps of about one molecule of water. Two/three peaks observed in the range of 200–330 °C relate to deprotonated steps of two dihydrogen phosphate (H_2_PO_4_^−^) anions. General mechanism reactions of thermal transformation could be:(2)$$\mathrm{Ca}(H_{2}{\mathrm{PO}}_{4}{)}_{2}\cdot H_{2}O(s)\overset{\mathrm{below}200{}^{\circ}C}{\to }\mathrm{Ca}(H_{2}{\mathrm{PO}}_{4}{)}_{2}(s)+H_{2}O(g)$$
(3)$$\mathrm{Ca}(H_{2}{\mathrm{PO}}_{4}{)}_{2}(s)\overset{200-330{}^{\circ}C}{\to }{\mathrm{Ca}}_{3}({\mathrm{PO}}_{3}{)}_{2}(s)+2H_{2}O(g)$$

The final decomposition of Ca(H_2_PO_4_)_2_·H_2_O to calcium polyphosphate Ca_3_(PO_3_)_2_, as revealed by reaction (3), occurred at above 400 °C. Many peaks in DTG curves observed for three Ca(H_2_PO_4_)_2_·H_2_O samples may indicate the splitting of each step of the dehydration step (reaction 2) and deprotonated hydrogen phosphate reaction (reaction 3). This result could be regarded to be affected by different inter/intramolecular interactions due to the different surroundings of water and H_2_PO_4_^−^ within the structure. Thermal properties of Ca(H_2_PO_4_)_2_·H_2_O prepared from different bivalve shells giving the different results indicate clearly that this property is dependent on raw materials.

Figure 7 presents FTIR spectra of the MCP-C, MCP-M, and MCP-O samples that are very similar because of fundamental vibrational bands of H_2_PO_4_^−^ and H_2_O block units in the Ca(H_2_PO_4_)_2_·H_2_O structure for each sample. The vibrational modes of the H_2_PO_4_^−^ anion are characterized by two types [47,48]: (i) the PO_4_^3−^ (T_d_ symmetry) internal vibrations, and (ii) the vibrations involving OH motions. The dihydrogen phosphate anion (H_2_PO_4_^−^) is a nonlinear structure containing seven atoms, which must have 15 (3 × 7 − 6) normal vibrational modes [43]. The nine vibrations coming from the PO_4_^3−^ (T_d_ symmetry) contain well-known normal modes: symmetric stretching (ν_1_(A_1_)), symmetric bending (ν_2_(E)), asymmetric stretching (ν_3_(F_2_)), and asymmetric bending (ν_4_(F_2_)) modes. The existence of two P-OH bonds results in a decreasing molecular symmetry of the H_2_PO_4_^−^ anion from its highest possible symmetry of the C_2v_ point group. As a result, the degenerate modes of ν_2_(E), ν_3_(F_2_), and ν_4_(F_2_) are fully lifted: ν_2_(E) separates into two modes (A_1_ + A_2_) and ν_3_(F_2_) and ν_4_(F_2_) into three modes (A_1_ + B_1_ + B_2_) each. These eight vibrations happen from the intra-ionic coupling interaction of two longer P- OH and two shorter P-O bonds for the PO_4_ stretching vibrations, which may also be led to additional modes as ν_s_(P(OH)_2_), ν_as_(P(OH)_2_), ν_s_(PO_2_), and ν_as_(PO_2_) for each H_2_PO_4_^−^ group. The six vibrations linking OH motions are characteristic for the H_2_PO_4_^−^ anion consisting of three modes (ν(OH), δ(OH), and γ(OH)) for each POH group. For water molecules, fundamental vibrations contain three normal vibrations: symmetric stretching (ν_1_(OH)), symmetric bending(ν_2_(HOH)), and asymmetric stretching (ν_3_(OH)) and three vibrations (wagging, rocking, and twisting). The bands observed in spectra of each sample are 493, 565, 690, 862, 963, 1091, 1164, 1237, 1388, 1679, 1700, 2311, 2440, 2960, 3263, 3470 cm^−1^, which are assigned to ν_2_(PO_4_^3−^), ν_4_(PO_4_^3−^), L_1_(H_2_O), γ(OH), ν_as_(P(OH)_2_), ν_s_(PO_2_), ν_as_(PO_2_), δ(OH) (1), δ(OH) (2), ν_2_(HOH), ν(OH) or C band, ν(OH) or B band, ν(OH) or B band, ν(OH) or A band, (ν_1_(OH)) of H_2_O, and (ν_3_(OH)) of H_2_O, respectively [49]. The FTIR results obtained are very similar to that of the Ca(H_2_PO_4_)_2_·H_2_O in literature [49], which confirms that the MCP-C, MCP-M, and MCP-O samples have major content as this crystal phase.

X-ray diffraction patterns of the MCP-C, MCP-M, and MCP-O samples (Figure 8) are the same 2θ positions but intense peaks are different. All detectable peaks of the obtained MCP-C, MCP-M, and MCP-O samples indexed as the Ca(H_2_PO_4_)_2_·H_2_O structure match with the standard data of PDF no. 70–0090 [41,46]. The XRD patterns exhibit two sharp characteristic peaks at 2θ = 22.95 and 24.18° corresponding to (0–21) and (210) reflections for anorthic crystal structure of Ca(H_2_PO_4_)_2_·H_2_O. The labeled diffraction peaks can be indexed according to standard XRD data and XRD peaks of other phases were not observed, confirming the pure compounds obtained under study. The XRD results and the FTIR data are well coincident.

Figure 9 presents the typical micrographs of the three selected powder (MCP-C, MCP-M, and MCP-O) samples. As shown in Figure 9, the MCP-C particles resemble polyhedral morphologies of sheet shapes with smooth surfaces. The MCP-M particles seem to have inherited the polyhedral morphologies of the plate-like microstructures with smooth surfaces. The MCP-O particles show polyhedral morphologies of lamellar-like shapes with smooth surfaces. Morphologies of three Ca(H_2_PO_4_)_2_·H_2_O samples prepared different bivalve (cockle, mussel, and oyster) shells are slightly different in shape and particle size but these morphologies are significantly from those of raw material powders.

### 3.3. Characterization Results of Tricalcium Phosphates

For FTIR spectra of tricalcium phosphates prepared from the calcination of the mixture of the produced monocalcium phosphates with its calcium raw materials (cockle, mussel, and oyster shells), labeled as the TCP-C, TCP-M, and TCP-O and shown in Figure 10. The FTIR spectra of each sample are very similar due to the fundamental vibrating unit, PO_4_^3−^ anion containing within the structure. Vibrational modes are discussed similarly with the previous section and 9 (3 × 5 − 6) normal vibrational modes of each phosphate group will be assigned [41]. In theory, the ν_3_(F_2_) and ν_4_(F_2_) modes are active in infrared while the ν_1_(A_1_) and ν_2_(E) modes are active in Raman [42]. Vibrational bands of the PO_4_^3−^ anion for all prepared products detected in the regions of 700–450 and 1250–900 cm^−1^ are defined to the ν_4_(PO_4_^3−^) and ν_3_(PO_4_^3−^) modes, respectively. Various vibrational peaks in these frequency regions insist on the presence of distinct nonequivalent phosphate block units in each structure and the loss of the degenerate modes of vibration resulting from correlation field splitting [42,43]. Additionally, a strong ν_s_(POP) band (721 cm^−1^) occurred is known to be the most striking characteristic of polyphosphate vibration.

The XRD patterns of TCP-C, TCP-M, and TCP-O samples were indexed as Ca_3_(PO_4_)_2_ structures, which are classified using the standard data from the International Centre for Diffraction Data (ICDD), exhibited in Figure 11. The powders performed by different shells match with PDF no. 55–0898. In all the prepared samples, the maximum peak relating with the crystallinity of beta phase was detected at 2θ = 31.5°, corresponding to the (021) planes, and indicating rhombohedral crystal phase, which agrees with the previous reports [50,51]. From the XRD data of the prepared TCP samples, no other identified peaks related to impurities and any intermediate or remaining raw materials are noted, which further insists on the purity of the synthesized β-Ca_3_(PO_4_)_2_ products. The obtained FTIR data of all the synthesized samples are consistent with the XRD results, which verify the classification of each compound.

The typical micrographs of the three selected powder (TCP-C, TCP-M, and TCP-O) samples are presented in Figure 12. The TCP-C particles look like the coarse surface on the grain-like shape and are highly agglomerated. The TCP-M particles were like polyhedral morphology of grain shape with uniform particles of 0.5–5.0 μm in size and agglomerations appear. The TCP-O particles were like polyhedral granular with identical particles of 0.2–5.0 μm in size and smooth surfaces. The difference of particle sizes and morphologies of three Ca_3_(PO_4_)_2_ samples was caused by the different kinds of used bivalve shells as raw reagents. The difference of particle sizes and morphologies of the three Ca_3_(PO_4_)_2_ samples was caused by the different kinds of used bivalve shells as raw reagents. The SEM micrograph data of the obtained products are significantly different from those of the raw material powders.

## 4. Conclusions

With depleting natural resources, it is important to find new sustainable sources of materials and in this research, MCP and TCP were successfully produced from bivalve shells that include cockle, mussel, and oyster. In addition, by recycling these seafood wastes we also help remove large amounts of shell wastes that could pose health and environmental hazards. The method reported herein is simple, rapid, cost-effective, and environmentally friendly. MCP and TCP obtained from this method showed resemble characteristics to previous reports, indicating high purities. The slight differences in thermal properties of the CaCO_3_ and MCP prepared depend on the starting shell powders. The morphologies of all the prepared samples were significantly different clearly indicating the raw material’s effect on this property. Overall, MCP and TCP converted from these bivalve shell wastes, have a huge potential to be used in many industrials bringing about both environmental and economic benefits.

## Figures and Tables

**Figure 1 materials-14-04395-f001:**
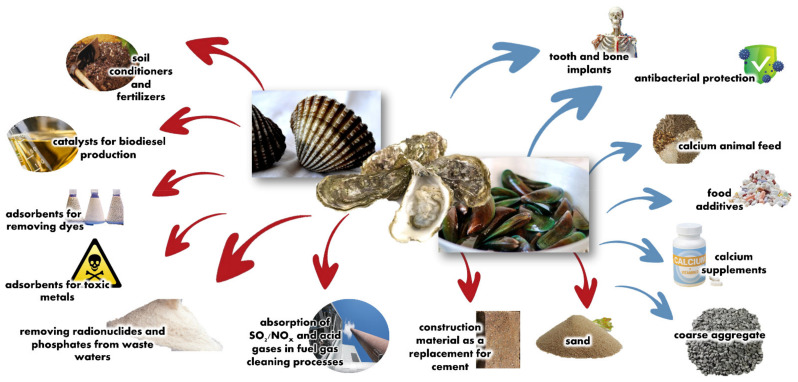
Potential applications of bivalve shell wastes.

**Figure 2 materials-14-04395-f002:**
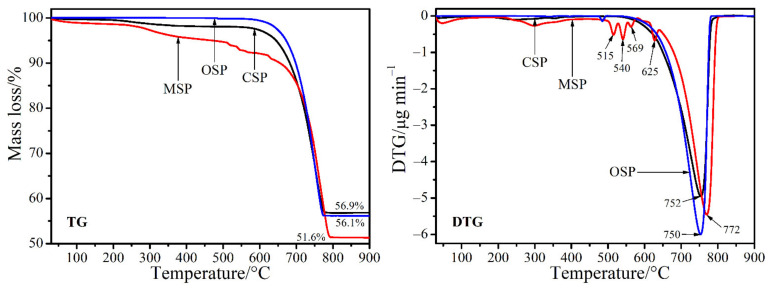
TG and DTG curves of the bivalve shell (CSP, MSP and OSP) powders.

**Figure 3 materials-14-04395-f003:**
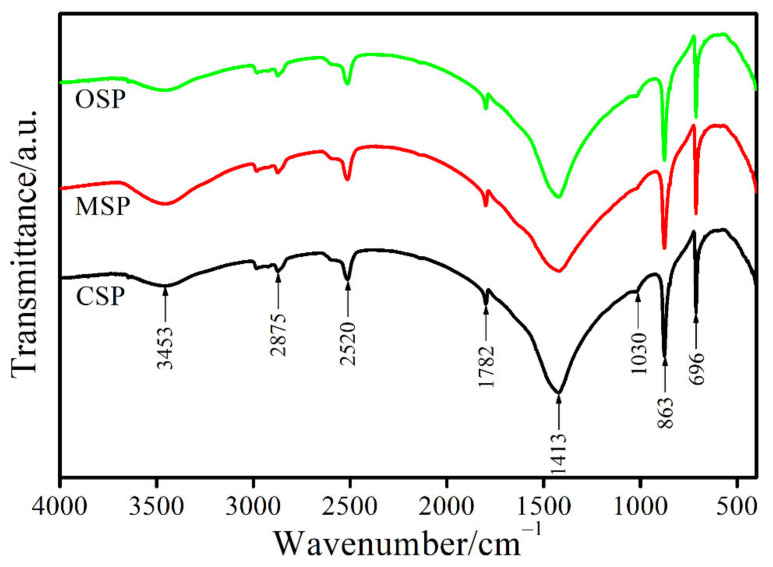
FTIR spectra of the bivalve shell (CSP, MSP and OSP) powders.

**Figure 4 materials-14-04395-f004:**
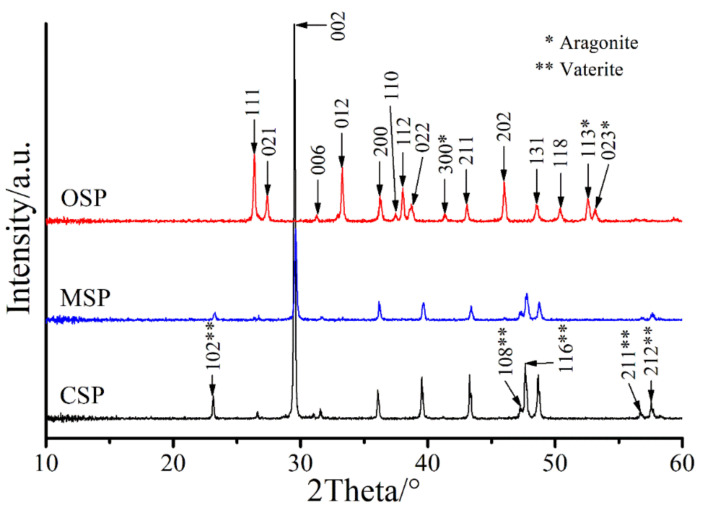
XRD patterns of the bivalve shell (CSP, MSP and OSP) powders.

**Figure 5 materials-14-04395-f005:**
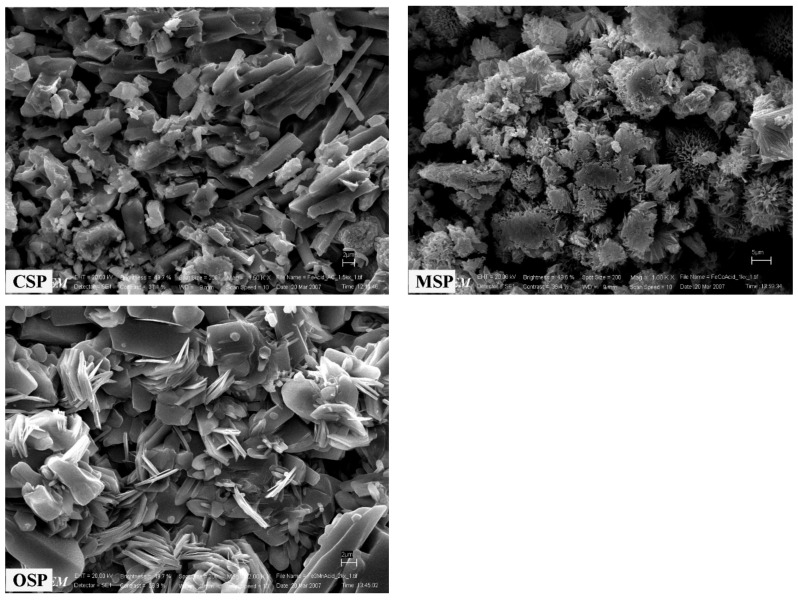
SEM micrographs of the bivalve shell (CSP, MSP and OSP) powders.

**Figure 6 materials-14-04395-f006:**
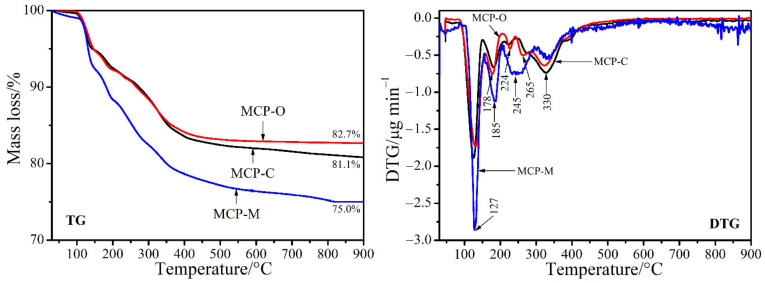
TG and DTG curves of MCP-C, MCP-M, and MCP-O samples.

**Figure 7 materials-14-04395-f007:**
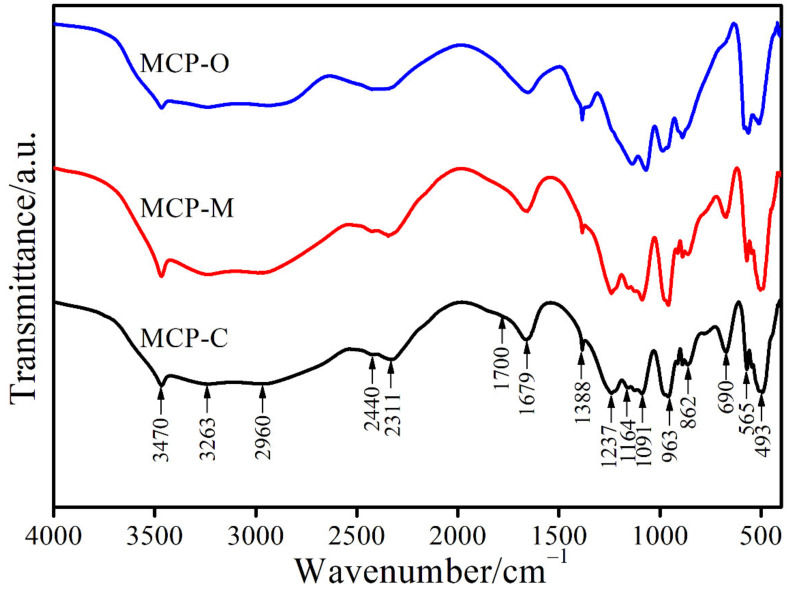
FTIR spectra of MCP-C, MCP-M, and MCP-O samples.

**Figure 8 materials-14-04395-f008:**
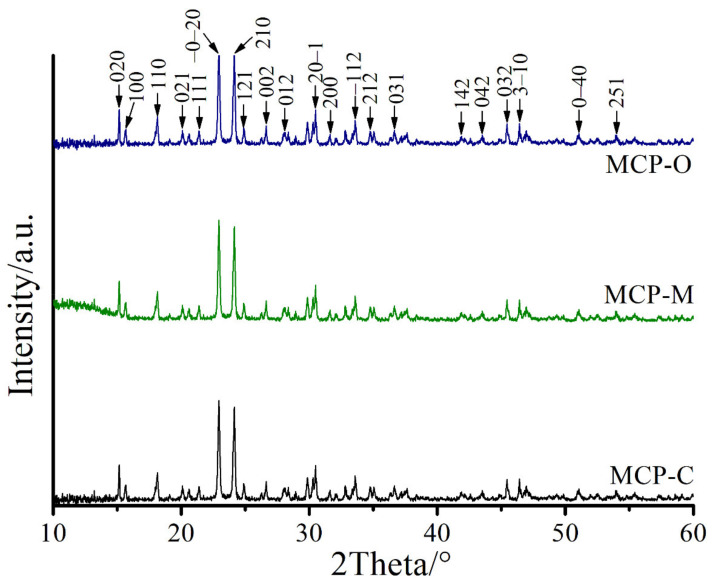
XRD patterns of MCP-C, MCP-M, and MCP-O samples.

**Figure 9 materials-14-04395-f009:**
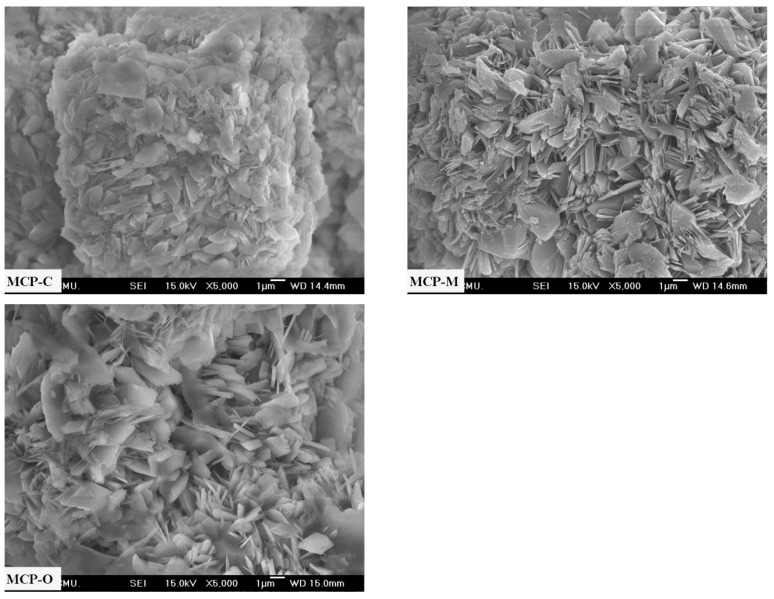
SEM micrographs of MCP-C, MCP-M and MCP-O samples.

**Figure 10 materials-14-04395-f010:**
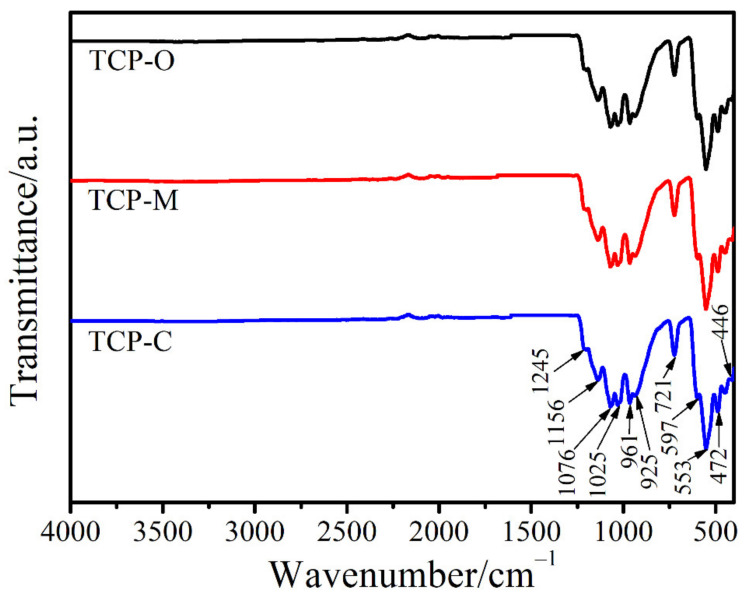
FTIR spectra of TCP-C, TCP-M and TCP-O samples.

**Figure 11 materials-14-04395-f011:**
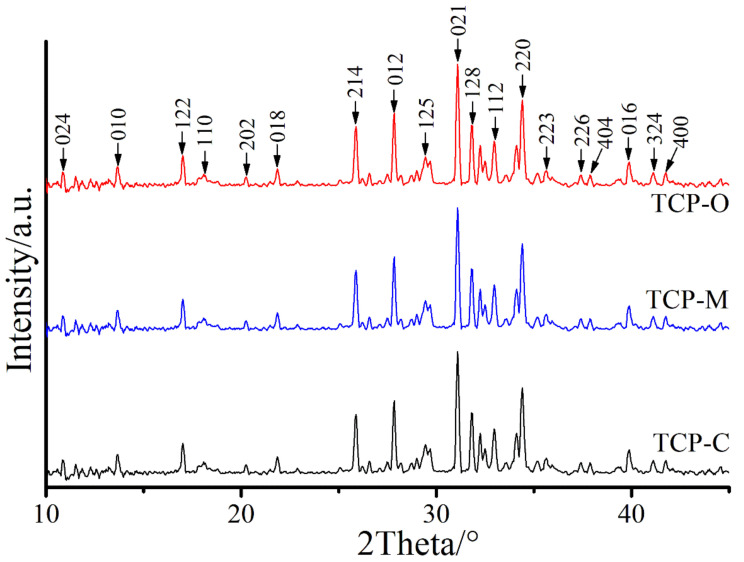
The XRD patterns of TCP-C, TCP-M, and TCP-O samples.

**Figure 12 materials-14-04395-f012:**
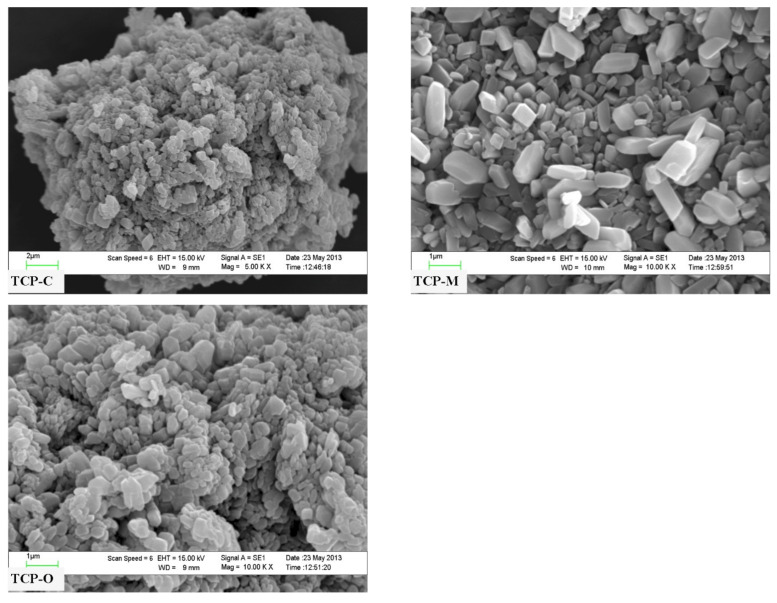
SEM micrographs of TCP-C, TCP-M and TCP-O samples.

## Data Availability

The data presented in this study are available on request from the corresponding author.

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
