# Peer review of "Conversion of Bivalve Shells to Monocalcium and Tricalcium Phosphates: An Approach to Recycle Seafood Wastes"

_materials, 2021, doi:10.3390/ma14164395_

Round 1

Reviewer 1 Report

Interesting written scientific paper, but some minor requirements are needed, as follows:

Please, be careful in the Abstract-the originality and novelty of research activity are not well exposed and/or highlighted.

The manuscript’s strengths are in the very atractive and important research field of conversion of bivalve shells to advanced calcium phosphate materials.

Weakness is in the future direction of your research. What is the application of your materials in the industry? The Bio-Circular-Green Economy (BCG) model is not well presented.

Generally, english language and style are fine, please check, some minor spells are required.

Author Response

Thank you for your good comments

Reviewer 2 Report

The work looks not very innovative to the reviewer.  The influences of minor elements of the raw materials on the results remain unclear. The authors should compare the results with pure reagents. The control group is missing. The author apparently should know pure CaCO3 is missing. Thus, I have to reject this study.

Author Response

Thank you for your good comments 

Reviewer 3 Report

The study is well-designed, and the manuscript is well-written. My only suggestion is to discuss how to environmentally friendly obtain enough 85% phosphoric acid. This is the key for the proposal to be used in production.

Author Response

Thank you for your good comments

Reviewer 4 Report

Dear authors,

The manuscript presents an interesting subject, but some improvements must be made to could be further considered for publication.

In this sense, relevant improvements are suggested in the manuscript:

- Title should be more objective and reduced.

- Some figures must be improved.

- Conclusion section must be rewritten.

The attached file presents (inside yellow sticky notes) the comments/suggestions that were annotated in the manuscript.

Author Response

Thank you for your good comments

Round 2

Reviewer 2 Report

I insist on the same comment: the manuscript should add pure CaCO3 as a control group...

Author Response

I insist on the same comment: the manuscript should add pure CaCO3 as a control group...

Response: Due to the COVID-19 situation, our university is closed for research. Therefore, we cannot redo the experiment with the addition of CaCO3 as a control group. In addition, we did cite related literature regarding the identity of CaCO3. We also checked other published papers that recycled seashells and they did not use pure CaCO3 as a control group (such as doi.org/10.12911/22998993/135316, doi.org/10.1016/j.jclepro.2020.123450, doi.org/10.1016/j.tca.2020.178568, doi:10.3390/ma11081410).

Reviewer 4 Report

Dear authors,

In general, the manuscript was improved, but persists several major (important and relevant) sentences that must be corrected.

In this sense, all the following comments are mandatory to be changed according:

- Line 23: In this case, grinding a CaCO3 raw material (seashells) do not transform the CaCO3 raw material in a new CaCO3 material. Grinding these raw materials also transform the raw materials in a powder form. Nothing else. In this sense, the line 23 or must be rewritten or is a fake scientific conclusion.

All this incorrect interpretation must be changed (this is mandatory) in all manuscript.

- Line 75: It is not just necessary change the text including “lime production”. Since references [7-10] do not have studied this subject, it is necessary to add (at least) one new and recent reference (this is mandatory) concerning that. For example: Eduardo Ferraz, José A F Gamelas, João Coroado, Carlos Monteiro & Fernando Rocha (2019) Recycling waste seashells to produce calcitic lime: characterization and wet slaking reactivity. Waste and Biomass Valorization, 10 (8): 2397-2414, https://doi.org/10.1007/s12649-018-0232-y

- Line 131: “100 mesh” (mesh is not SI unit) corresponds to “150 µm”, not to “0-150 µm”.

- Line 223: As stated in previous review, how was obtained this semiquantification “≥ 96 wt%” by XRD? It must be written in the “Materials and methods” XRD section the method/technic concerning this subject. This is mandatory.

What is the meaning of “little”? What is the meaning of “compositions”?

- Figure 4: The interpretation of the XRD data must be made careful and accurate (not to dispatch the review).

How the authors justify the identification of vaterite 116 peak (with an intensity peak around 50), if the must relevant vaterite peaks (100 intensity): 110, 111, 112, 300 and 114 are not presented in the pattern? How the authors justify the identification of aragonite 113 (25 intensity) peak, if the must relevant aragonite peaks: 111 (100 intensity), 211 (65 intensity), 021 (52 intensity), 021 (46 intensity) are not presented in the pattern? This is impossible in XRD.

The reported PDF 72-1937 file for calcite identification (line 230 of the manuscript) do not identify the main peak (around 29.4 2theta, d = 3.04) as 002 hkl (as reported in figure), but as 104. Several hkl identifications are incorrect. Please, correct all the hkl presented in figure by the correct hkl presented in the referred PDF 72-1937.

There are several articles where the identification of the phases presented in the studied seashell is correct. It is mandatory correct this important subject.

- Line 225: “Each sample”? Aragonite also occurs in oyster and vaterite also in the other two. This sentence must be corrected concerning the correct XRD identification.

- Line 230: It is not correct, since XRD results do not well agree with FTIR data. How the authors justify the identification of aragonite and vaterite by XRD, and these phases were not detected by FTIR which is a more precise and accurate technic? If the phases are present, it is impossible do not detect aragonite and vaterite by FTIR!

- Lines 249-250: As stated in previous review, this is not correct. The total mass for MCP-M is 25%. 25% is completely different from 21%. The explanation and interpretation provided by the authors “MCP should be Ca(H2PO4)2.1.2H2O. This is possible and has been reported in previous work” must be include in text, as well as the respective reference to previous work. This is mandatory.

- Figure 6 DTG: The figure was not improved. Several peaks (178, 224 and 265) are not assigned to the respective peak (with an arrow, for example). This is mandatory.

- Lines 261-262: Please, rewrite this sentence in English: The final decomposition of Ca(H2PO4)2•H2O to calcium polyphosphate Ca3(PO3)2, as revealed by reaction (3), occurred above 600 oC”. This is mandatory.

Why 600 ºC? In the previous version was referred 400 ºC, which is according the DTG results, where the reaction is “completed” above 400-450 ºC.

- Line 355: As stated in previous review, “grain rice-like” is not a morphological designation, maybe “fusiform-like”. A new confusion was created with the “polyhedral-like” designation. Lamellar or plate is a polyhedral morphology. What is the difference from “polyhedral” to lamellar or plate? Check and change this in all manuscript. This is mandatory.

- Line 356: What is the meaning of “0.5.0”?

- Line 340: As stated in previous review, ”Joint Committee on Pow-340 der Diffraction and Standards (JCPDS)” is the former designation. Change to the actual designation “International Centre for Diffraction Data (ICDD)”. This is mandatory.

- As stated in previous review, the figures Fig. 2 TG and DTG, Fig. 3, Fig. 4, Fig. 6 TG and DTG, Fig. 7, Fig. 8 have the legend of the x and y axes with not uniformity format.

Example: “Transmitance/ a.u.” is different from “Transmitance/a.u.” and from “Transmitance / a.u.” and from “Transmitance /a.u.”. Please, change all the legends to the same uniformity.

Author Response

Dear authors,

In general, the manuscript was improved, but persists several major (important and relevant) sentences that must be corrected.

In this sense, all the following comments are mandatory to be changed according:

- Line 23: In this case, grinding a CaCO3 raw material (seashells) do not transform the CaCO3 raw material in a new CaCO3 material. Grinding these raw materials also transform the raw materials in a powder form. Nothing else. In this sense, the line 23 or must be rewritten or is a fake scientific conclusion.

All this incorrect interpretation must be changed (this is mandatory) in all manuscript.

Response: The sentence we edited to in the abstract was “By grinding, calcium carbonate (CaCO3) powder was obtained from each shell of bivalves (cockle, mussel, and oyster) as revealed by FTIR and XRD results.” We used the term “obtained” and not “transformed”. So, we did say that the power was obtained and not transformed from shells. The original manuscript used the word “transformed” (By grinding, each shell of bivalves (cockle, mussel, and oyster) was transformed to the same crystal type of calcite phase of CaCO3, revealed by FTIR and XRD results.) There is no other place in the manuscript that we stated that the shells were transformed to CaCO3.

- Line 75: It is not just necessary change the text including “lime production”. Since references [7-10] do not have studied this subject, it is necessary to add (at least) one new and recent reference (this is mandatory) concerning that. For example: Eduardo Ferraz, José A F Gamelas, João Coroado, Carlos Monteiro & Fernando Rocha (2019) Recycling waste seashells to produce calcitic lime: characterization and wet slaking reactivity. Waste and Biomass Valorization, 10 (8): 2397-2414, https://doi.org/10.1007/s12649-018-0232-y

Response: We have edited this part and added 12 new recent references including the suggested one.

- Line 131: “100 mesh” (mesh is not SI unit) corresponds to “150 µm”, not to “0-150 µm”.

Response: This has been edited.

- Line 223: As stated in previous review, how was obtained this semi quantification “≥ 96 wt%” by XRD? It must be written in the “Materials and methods” XRD section the method/technic concerning this subject. This is mandatory.

What is the meaning of “little”? What is the meaning of “compositions”?

Response:  compositions meanings chemical contents

- Figure 4: The interpretation of the XRD data must be made careful and accurate (not to dispatch the review).

How the authors justify the identification of vaterite 116 peak (with an intensity peak around 50), if the must relevant vaterite peaks (100 intensity): 110, 111, 112, 300 and 114 are not presented in the pattern? How the authors justify the identification of aragonite 113 (25 intensity) peak, if the must relevant aragonite peaks: 111 (100 intensity), 211 (65 intensity), 021 (52 intensity), 021 (46 intensity) are not presented in the pattern? This is impossible in XRD.

The reported PDF 72-1937 file for calcite identification (line 230 of the manuscript) do not identify the main peak (around 29.4 2theta, d = 3.04) as 002 hkl (as reported in figure), but as 104. Several hkl identifications are incorrect. Please, correct all the hkl presented in figure by the correct hkl presented in the referred PDF 72-1937.

There are several articles where the identification of the phases presented in the studied seashell is correct. It is mandatory correct this important subject.

Response:  the authors don’t agree with the reviewer and the XRD information of CaCO3 obtained from seashells had sufficiently reported. 

- Line 225: “Each sample”? Aragonite also occurs in oyster and vaterite also in the other two. This sentence must be corrected concerning the correct XRD identification.

Response:  It can clearly indicate that the main crystalline phases of the CSP, MSP, and OSP samples are calcite (β-CaCO3) and minor aragonite are detected for OSP and little vaterite are detected for other two samples, shown in Figure 4.

- Line 230: It is not correct, since XRD results do not well agree with FTIR data. How the authors justify the identification of aragonite and vaterite by XRD, and these phases were not detected by FTIR which is a more precise and accurate technic? If the phases are present, it is impossible do not detect aragonite and vaterite by FTIR!

Response: The authors don’t agree with the reviewer because all phases of calcium carbonate (aragonite, vaterite and calcite) contain same CO32- block unit, which has detected similar fundamental vibration for FTIR technique. Dominant vibrational characterization of each phase may be not observed because of many reasons such as minor phases in the solid or/and overlapping vibrational bands of the main phase.  The XRD technique is a primary tool for the characterization of phase of solid, which confirms the main calcite phase with minor phases of aragonite and vaterite.  In our opinion, it is correct that XRD results do well agree with FTIR data.

- Lines 249-250: As stated in previous review, this is not correct. The total mass for MCP-M is 25%. 25% is completely different from 21%. The explanation and interpretation provided by the authors “MCP should be Ca(H2PO4)2.1.2H2O. This is possible and has been reported in previous work” must be include in text, as well as the respective reference to previous work. This is mandatory.

Response:  This has been edited as follow:

 Total mass losses found to be 18.9% for MCP-C, 25.0% for MCP-M and 17.3% for MCP-O are slightly different from that of theoretical values at 21.4 % [42, 47].  The obtained results indicate that the number of water molecules in Ca(H2PO4)2•nH2O structure would be 0.7, 1.5 and 0.4 mole for the MCP-C, MCP-M, and MCP-O samples, respectively.  The number of crystal water found in the range of 0-n<2 is impossible theory, which is consistent with that of the previous works [24, 26, 42, 47].

- Figure 6 DTG: The figure was not improved. Several peaks (178, 224 and 265) are not assigned to the respective peak (with an arrow, for example). This is mandatory.

Response: This has been edited.

- Lines 261-262: Please, rewrite this sentence in English: The final decomposition of Ca(H2PO4)2•H2O to calcium polyphosphate Ca3(PO3)2, as revealed by reaction (3), occurred above 600 oC”. This is mandatory.

Why 600 ºC? In the previous version was referred 400 ºC, which is according the DTG results, where the reaction is “completed” above 400-450 ºC.

Response: This has been edited.

- Line 355: As stated in previous review, “grain rice-like” is not a morphological designation, maybe “fusiform-like”. A new confusion was created with the “polyhedral-like” designation. Lamellar or plate is a polyhedral morphology. What is the difference from “polyhedral” to lamellar or plate? Check and change this in all manuscript. This is mandatory.

Response:  This has been edited. However, the difference from “polyhedral” to lamellar or plate depends on the viewpoints of each researcher.

- Line 356: What is the meaning of “0.5.0”?

Response: This was a typo and has been changed to 0.5.

- Line 340: As stated in previous review, ”Joint Committee on Pow-340 der Diffraction and Standards (JCPDS)” is the former designation. Change to the actual designation “International Centre for Diffraction Data (ICDD)”. This is mandatory.

Response: This has been edited.

- As stated in previous review, the figures Fig. 2 TG and DTG, Fig. 3, Fig. 4, Fig. 6 TG and DTG, Fig. 7, Fig. 8 have the legend of the x and y axes with not uniformity format.

Example: “Transmitance/ a.u.” is different from “Transmitance/a.u.” and from “Transmitance / a.u.” and from “Transmitance /a.u.”. Please, change all the legends to the same uniformity.

Response: All graphs have been reformatted for uniformity.